# HUMAN-INTERPRETABLE MODEL EXPLAINABILITY ON HIGH-DIMENSIONAL DATA

## ABSTRACT

The importance of explainability in machine learning continues to grow, as both neural-network architectures and the data they model become increasingly complex. Unique challenges arise when a model's input features become high dimensional: on one hand, principled model-agnostic approaches to explainability become too computationally expensive; on the other, more efficient explainability algorithms lack natural interpretations for general users. In this work, we introduce a framework for human-interpretable explainability on high-dimensional data, consisting of two modules. First, we apply a semantically-meaningful latent representation, both to reduce the raw dimensionality of the data, and to ensure its human interpretability. These latent features can be learnt, e.g. explicitly as disentangled representations or implicitly through image-to-image translation, or they can be based on any computable quantities the user chooses. Second, we adapt the Shapley paradigm for model-agnostic explainability to operate on these latent features. This leads to interpretable model explanations that are both theoretically-controlled and computationally-tractable. We benchmark our approach on synthetic data and demonstrate its effectiveness on several image-classification tasks.

## 1 INTRODUCTION

The explainability of AI systems is important, both for model development and model assurance. This importance continues to rise as AI models – and the data on which they are trained – become ever more complex. Moreover, methods for AI explainability must be adapted to maintain the human-interpretability of explanations in the regime of highly complex data.

Many explainability methods exist in the literature. Model-specific techniques refer to the internal structure of a model in formulating explanations (Chen & Guestrin, 2016; Shrikumar et al., 2017), while model-agnostic methods are based solely on input-output relationships and treat the model as a black-box (Breiman, 2001; Ribeiro et al., 2016). Model-agnostic methods offer wide applicability and, importantly, fix a common language for explanations across different model types.

The Shapley framework for model-agnostic explainability stands out, due to its theoretically principled foundation and incorporation of interaction effects between the data's features (Shapley, 1953; Lundberg & Lee, 2017). The Shapley framework has been used for explainability in machine learning for years (Lipovetsky & Conklin, 2001; Kononenko et al., 2010; Štrumbelj & Kononenko, 2014; Datta et al., 2016). Unfortunately, the combinatorics required to capture interaction effects make Shapley values computationally intensive and thus ill-suited for high-dimensional data.

More computationally-efficient methods have been developed to explain model predictions on high-dimensional data. Gradient- and perturbation-based methods measure a model prediction's sensitivity to each of its raw input features (Selvaraju et al., 2020; Zhou et al., 2016; Zintgraf et al., 2017). Other methods estimate the mutual information between input features and the model's prediction (Chen et al., 2018a; Schulz et al., 2020), or generate counterfactual feature values that change the model's prediction (Chang et al., 2019; Goyal et al., 2019; Wang & Vasconcelos, 2020). See Fig. 1 for explanations produced by several of these methods (with details given in Sec. 3.5).

When intricately understood by the practitioner, these methods for model explainability can be useful, e.g. for model development. However, many alternative methods exist to achieve broadly the same goal (i.e. to monitor how outputs change as inputs vary) with alternative design choices that

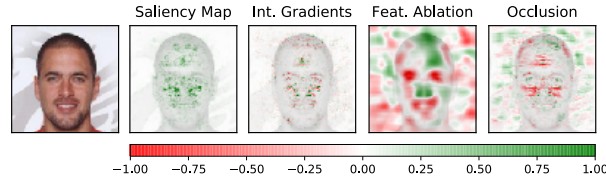

Figure 1: Pixel-based explanations of a model trained to predict the attractiveness label in CelebA.

Figure 2: Our proposed framework for semantic explainability.

make their explanations uncomparable to a general user: e.g. the distinct explanations in Fig. 1 describe the same model prediction. Ideally, a set of axioms (agreed upon or debated) would constrain the space of explanations, thus leading to a framework of curated methods that the user can choose from based on which axioms are relevant to the application.

A further challenge on high-dimensional data is the sheer complexity of an explanation: in the methods described above, explanations have the same dimensionality as the data itself. Moreover, the importance of raw input features (e.g. pixels) are not individually meaningful to the user. Even when structured patterns emerge in an explanation (e.g. in Fig. 1) this is not sufficient to answer higher-level questions. For example, did the subject's protected attributes (e.g. gender, age, or ethnicity) have any influence on the model's decision?

In this work, we develop methods for explaining predictions in terms of a digestible number of semantically meaningful concepts. We provide several options for transforming from the high-dimensional raw features to a lower-dimensional latent space, which allow varying levels of user control. Regardless of the method used, transformation to a low-dimensional human-interpretable basis is a useful step, if explanations are to satisfy experts and non-experts alike.

Once a set of semantic latent features is selected, one must choose an explainability algorithm to obtain quantitative information about why a certain model prediction was made. Fortunately, since the set of latent features is low-dimensional by construction, a Shapley-based approach becomes once again viable. In this work, we develop a method to apply Shapley explainability at the level of semantic latent features, thus providing a theoretically-controlled, model-agnostic foundation for explainability on high-dimensional data. Our main contributions are:

- We introduce an approach to model explainability on high-dimensional data that involves encoding the raw input features into a digestible number of semantically meaningful latent features. We develop a procedure to apply Shapley explainability in this context, obtaining Shapley values that describe the high-dimensional model's dependence on each semantic latent feature.

- We demonstrate 3 methods to extract semantic features for the explanations: Fourier transforms, disentangled representations, and image-to-image translation. We benchmark our approach on dSprites – with known latent space – and showcase its effectiveness in computer vision tasks such as MNIST, CIFAR-10, ImageNet, Describable Textures, and CelebA.

## 2  SEMANTIC SHAPLEY EXPLAINABILITY

In this section, we present a simple modular framework for obtaining meaningful low-dimensional explanations of model predictions on high-dimensional data. The framework contains two modules: (i) a mechanism for transforming from the high-dimensional space of raw model inputs to a low-dimensional space of semantic latent features, and (ii) an algorithm for generating explanations of the model's predictions in terms of these semantic features. See Fig. 2.

We will begin by describing module (ii) in Sec. 2.1, where we will show how to adapt Shapley explainability to latent features. Then we will describe several options for module (i) in Sec. 2.2.

### 2.1  SHAPLEY VALUES FOR LATENT FEATURES

Shapley values (Shapley, 1953) were developed in cooperative game theory to distribute the value $v(N)$ earned by a team $N = \{1, 2, \ldots, n\}$ among its players. The Shapley value $\phi_v(i)$ represents

the marginal value added by player $i$ upon joining the team, averaged over all orderings in which the team can form. In particular,

$$\phi_v(i) = \sum_{S \subseteq N \setminus \{i\}} \frac{|S|! \, (n - |S| - 1)!}{n!} \big[ v(S \cup \{i\}) - v(S) \big] \tag{1}$$

where $v(S)$ represents the value that a coalition $S$ obtains without the rest of their teammates. Shapley values are the unique attribution method satisfying 4 natural axioms (Shapley, 1953). For example, they sum to the total value earned: $\sum_i \phi_v(i) = v(N) - v(\{\})$, and they are symmetric if two players $i$ and $j$ are functionally interchangeable: $\phi_v(i) = \phi_v(j)$. Shapley values thus serve as a well-founded explanation of an output (the earned value) in terms of inputs (the players).

The method can be adapted to explain the output of a machine learning model by interpreting the model's input features $x = (x_1, \ldots, x_n)$ as the players of a game. Consider a classification task, and let $f_y(x)$ be the model's predicted probability that data point $x$ belongs to class $y$. To apply Shapley explainability, one must define a value function representing the model's expected output given only a subset of the input features $x_S$. The most common choice is

$$v_{f_y(x)}(S) = \mathbb{E}_{p(x')} \big[ f_y(x_S \sqcup x'_{\bar{S}}) \big] \tag{2}$$

where $p(x')$ is the distribution from which the data is drawn, $\bar{S}$ is the complement of $S$, and $x_S \sqcup x'_{\bar{S}}$ represents the spliced data point with in-coalition features from $x$ and out-of-coalition features from $x'$. Then, inserting the value function of Eq. (2) into the definition of Eq. (1), one obtains Shapley values $\phi_{f_y(x)}(i)$ representing the portion of the prediction $f_y(x)$ attributable to feature $x_i$.

The Shapley values presented above provide a *local* explanation of the model's behaviour on an individual data point. For a *global* explanation, local values can be aggregated (Frye et al., 2020)

$$\Phi_f(i) = \mathbb{E}_{p(x,y)} \big[ \phi_{f_y(x)}(i) \big] \tag{3}$$

where $p(x, y)$ is the joint distribution from which the labelled data is drawn. This aggregation preserves the Shapley axioms and is motivated by the sum rule:

$$\sum_i \Phi_f(i) = \mathbb{E}_{p(x,y)} \big[ f_y(x) \big] - \mathbb{E}_{p(x')p(y)} \big[ f_y(x') \big] \tag{4}$$

which can be interpreted as the model accuracy above a class-balance baseline. Global Shapley values thus represent the portion of model accuracy attributable to each feature.

To adapt the Shapley framework to latent features, suppose (as in Fig. 2) that a mapping $x \to z$ exists to transform the raw model inputs $x$ into a semantically meaningful representation $z(x)$, and that an (approximate) inverse mapping $z \to \tilde{x}$ exists as well. Then we can obtain an explanation of the model prediction $f_y(x)$ in terms of the latent features $z(x)$ by applying Shapley explainability instead to the function $f_y(\tilde{x}(z))$ at the point $z = z(x)$. To be precise, we define a value function

$$\tilde{v}_{f_y(x)}(S) = \mathbb{E}_{p(x')} \left[ f_y \Big( \tilde{x} \big( z_S(x) \sqcup z_{\bar{S}}(x') \big) \Big) \right] \tag{5}$$

which represents the marginalisation of $f_y(\tilde{x}(z))$ over out-of-coalition features $z_{\bar{S}}$. Here $z_S(x)$ is the in-coalition slice of $z(x)$, and $z_{\bar{S}}(x')$ is the out-of-coalition slice corresponding to a different data point. These get spliced together in latent space before transforming back to model-input-space and feeding into the model. Inserting the value function of Eq. (5) into the definition of Eq. (1) produces semantic Shapley values that explain $f_y(x)$ in terms of latent features $z_i$.

## 2.2 LANDSCAPE OF SEMANTIC REPRESENTATIONS

A wide variety of methods exist to transform from the high-dimensional set of raw model inputs to an alternative set of features that offer semantic insight into the data being modelled. In this section, we consider several options for this semantic component of our approach to explainability.

### 2.2.1 FOURIER TRANSFORMS

Despite their high dimensionality, pixel-based explanations as in Fig. 1 manage to convey meaning to their consumers through the local structure of images. A central claim of our work, however, is

that such meaning remains limited and incomplete. One way to complement the location information of pixel-based explanations is with frequency-based explanations via Fourier transforms.

If an image consists of a value $x_i$ for each pixel $i = 1, \ldots, n$, then its discrete Fourier transform (Cooley & Tukey, 1965) consists of a value $z_k$ for each Fourier mode $k = 1, \ldots, n$. Each mode $k$ corresponds to a set of frequencies in the horizontal and vertical directions, ranging from 0 (uniform imagery) to $1/2$ (one oscillation every 2 pixels). As the inverse Fourier transform $\tilde{x}(z(x)) = x$ exists, we can obtain Shapley values for the Fourier modes of an image using Eq. (5).

We can also aggregate Fourier-based Shapley values into frequency bins to reduce the complexity of the calculation and lower the dimensionality of the explanation. Such frequency-based explanations can offer insight into whether a model's decisions are based on shapes or textures (see Sec. 3.2) and whether a model is robust to adversarial examples (see Sec. 3.1).

### 2.2.2 DISENTANGLED REPRESENTATIONS

The goal of disentangled representation learning (Bengio et al., 2013; Schmidhuber, 1991) is to learn a mapping from a data set's raw high-dimensional features to a lower-dimensional basis of semantically meaningful factors of variation. In an image, these factors of variation might correspond to the subject's position and orientation, their emotional state, the lighting, and the setting. Disentangled representations are highly aligned with our goal of achieving semantic explanations.

Different approaches offer varying levels of control over which semantic features are learnt in the representation. Unsupervised disentanglement extracts factors of variation directly from the data. Such methods are often based on variational inference (Higgins et al., 2017; Burgess et al., 2018; Kim & Mnih, 2018; Chen et al., 2018b) – though other approaches exist (Chen et al., 2016) – and often seek a factorised latent representation. Supervised disentanglement (Schmidhuber, 1991; Lample et al., 2017) allows the practitioner to specify a subset of the sought-after semantic features by labelling them – in full or in part (Locatello et al., 2020) – in the data. Such methods then involve learning a representation that factorises the specified and unspecified factors of variation.

Disentangled representations based on variational autoencoders (Kingma & Welling, 2014) include an encoder $z(x)$ and a decoder $\tilde{x}(z)$, thus fitting neatly into our framework (Fig. 2) for semantic explainability. The value function of Eq. (5) then leads to Shapley values that explain a model's predictions in terms of the disentangled factors of variation underlying the data. We will demonstrate this for unsupervised (Sec. 3.3) and supervised (Sec. 3.4) disentanglement with experiments.

### 2.2.3 IMAGE-TO-IMAGE TRANSLATION

In image-to-image translation (Isola et al., 2017), one is not interested in directly extracting semantic factors of variation, but instead in transforming images by selectively modifying underlying semantic features. Generative adversarial methods can accomplish this goal without passing through an explicit compressed representation (Zhu et al., 2017). Other adversarial methods allow the user to selectively perturb the semantic attributes of an image (e.g. hair colour or gender) for attributes that are labelled in the data set (Choi et al., 2018; He et al., 2019; Liu et al., 2019a).

Image-to-image translation methods can straightforwardly be incorporated into our framework for semantic explainability. To do so, one inserts the result of the translation $x \to \tilde{x}\big(z_S(x) \sqcup z_{\bar{S}}(x')\big)$, which corresponds to the modification of semantic attributes $z_{\bar{S}}(x) \to z_{\bar{S}}(x')$, into the value function of Eq. (5). We demonstrate this in Sec. 3.5 below.

## 3 RESULTS

Here we demonstrate the practical utility of our approach on a variety of data sets and for a diverse set of semantic representations. We use pre-trained models for the semantic representations to show that our method can be applied with existing tools. See App. A for full experimental details.

### 3.1 CIFAR-10

We begin by applying our explainability framework using the Fourier transform as the semantic representation. We do this on CIFAR-10 (Krizhevsky et al., 2009) and investigate whether sensitivity

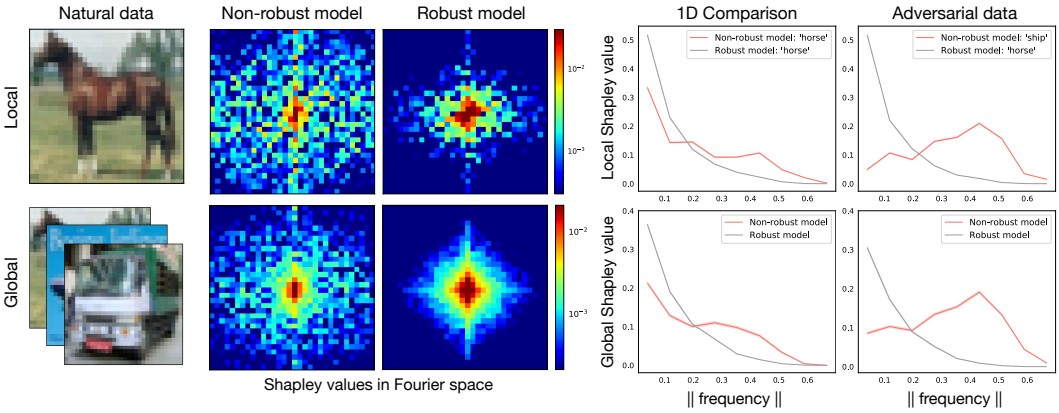

Figure 3: Fourier-space model explanations on CIFAR-10 shedding light on adversarial sensitivity.

to adversarial examples (Szegedy et al., 2014; Goodfellow et al., 2015; Nguyen et al., 2015) is linked to dependence on high-frequency (i.e. small length-scale) fluctuations. We considered two classifiers on CIFAR-10: a robust model (Madry et al., 2018; Engstrom et al., 2019) trained to be insensitive to adversarial perturbations, and a non-robust model (He et al., 2016) trained naturally. We computed semantic Shapley values according to Sec. 2.2.1.

The first row of Fig. 3 shows the semantic explanations that result for a single image. The Shapley values in Fourier space show that, on this particular image, the non-robust model is sensitive to high-frequency features (less detectable by the human eye), while the robust prediction is based exclusively on low-frequency information near the Fourier-space origin (large-scale structure). The 1-dimensional comparison, in which frequency modes were binned according to Euclidean norm, allows for quantitative comparison that confirms this effect.

We also computed the adversarial perturbation of the local image in Fig. 3, using projected gradient descent (Nocedal & Wright, 2006) and $\epsilon = 8/255$ (separately for each model). Such adversarial perturbations do not significantly alter the image to the human eye, and the robust model's prediction accordingly remains unchanged; however, the non-robust model's prediction is perturbed from "horse" to "ship". Fig. 3 explains the model decisions on these adversarially perturbed images, showing that the non-robust model's mistake is due to high-frequency features.

The second row of Fig. 3 shows global semantic explanations for these models, which correspond to aggregating local explanations across the data set. We see that the trends found above for one particular image hold in general throughout the data. Our framework for semantic explainability thus leads to the interesting result that adversarially robust classifiers are less sensitive to high-frequency information. See App. A.2 for similar results on ImageNet (Deng et al., 2009).

## 3.2 DESCRIBABLE TEXTURES

Here we apply our framework – again using the Fourier transform – to explain whether a model's predictions are based on shapes or textures. Shapes tend to correspond to extended objects and thus sit at the lower end of the frequency spectrum. Textures correspond to small-scale patterns and thus occupy the higher end of the frequency spectrum, often with distinctive peaks that represent a periodic structure. We explore this question by explaining a ResNet-50 (He et al., 2016) trained on the Describable Textures Dataset (Cimpoi et al., 2014).

Fig. 4 shows randomly drawn "banded" and "woven" images from the data set, as well as their pixel-based explanations computed with integrated gradients (Sundararajan et al., 2017). At the level of pixels, it is difficult to judge what qualities (e.g. colours, shapes, textures) drive the model's prediction. However, the frequency-based explanations in Fig. 4 show clear peaks at high frequencies corresponding to regular patterns in each image. See App. A.3 for additional examples.

Figure 4: Frequency-based explanations on Describable Textures, showing sensitivity to periodicity.

### 3.3 DSPRITES

Here we apply our explainability framework using unsupervised disentangled representations to extract the semantic features. We do this on dSprites (Matthey et al., 2017), synthetic images of sprites (see Fig. 5a) generated from a known latent space with 5 dimensions: shape, scale, orientation, and horizontal and vertical positions. This experiment serves as a benchmark of our approach.

In this experiment, we explain a rules-based model that classifies sprites according to the number of white pixels in the top half of the image (with 6 classes corresponding to 6 latent scales). We articulate the explanation in terms of the unsupervised disentangled representation (cf. Sec. 2.2.2) of a $\beta$-TCVAE (Chen et al., 2018b), using a publicly available model (Dubois, 2019).

Fig. 5a shows semantic Shapley values for this model. Globally, the model relies on the sprite's vertical position and scale, while generally ignoring its shape, orientation, and horizontal position. This is indeed consistent with the rules-based model we laid out above. Locally, the model classifies the sprite as $y = 0$, as it has no white pixels in the top half of the image. The vertical position of the sprite is the main driver of this decision. In fact, the sprite's scale (maximum) would tend to indicate a different class, so the scale receives a negative local Shapley value in this case.

This experiment validates our framework for semantic explainability, as the modelling task and factors of variation in the data are fully understood and consistent with our results. Moreover, this example showcases an explanation that differentiates between shape and scale – semantic features that cannot be distinguished in a pixel-based explanation.

### 3.4 MNIST

Here we apply our framework – again using disentangled representations – to a binary classifier on MNIST (LeCun & Cortes, 2010). In particular, we explain a rules-based model that detects the presence of a geometric hole, which exists when all black pixels are not contiguous. We use the unsupervised disentangled representation of a JointVAE pre-trained on MNIST (Dupont, 2018), which accommodates a combination of continuous (e.g. handwriting) and discrete (e.g. digit value) latent dimensions. Supervised methods also exist for disentangling the digit value in the latent representation (Makhzani et al., 2016).

Fig. 5b shows semantic Shapley values for the rules-based model. The global values show that the digit value has the most bearing on the model's general behaviour; indeed, some digits (0, 6, 8, 9) almost always contain a geometric hole while others (1, 3, 5, 7) almost always do not. The global values also show sensitivity to writing style (e.g. 4 vs $\mathit{4}$) and stroke thickness (which can close a hole) as expected. The local values in Fig. 5b are roughly in line with the global trends.

This example demonstrates that latent disentanglement is a powerful tool for understanding model decisions at a semantic level. These results also highlight that explanations are only as good as the chosen semantic representation. One can see in the latent traversals of Fig. 5b that in this representation, semantic concepts are not perfectly disentangled: stroke thickness and hole size are entangled, and writing style mixes 7's and 9's. This can lead to a small corruption of the explanation.

### 3.5 CELEBA

Here we apply our explainability framework using image-to-image translation as the implicit semantic representation. We do this on CelebA (Liu et al., 2015) to demonstrate, on real world data,

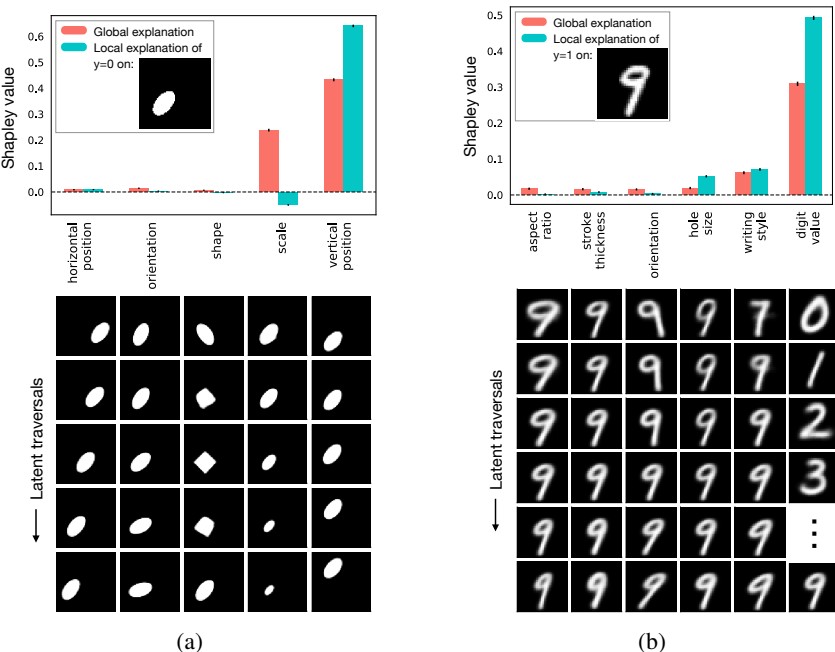

Figure 5: Model explanations in terms of disentangled latent features on (a) dSprites and (b) MNIST.

that our method elucidates patterns in model behaviour that are missed by other methods. In particular, we train a CNN to predict the labelled attractiveness of individuals in the data. We choose this admittedly banal label because it is influenced by a variety of other higher level attributes, including sensitive characteristics (e.g. gender, age, skin tone) that should not influence fair model predictions. Pixel-based explanations (e.g. Fig. 1) provide no insight into these important issues.

We explain the model's predictions in terms of other semantic attributes also labelled in the data. We do so according to Sec. 2.2.3, using the implicit representation of an STGAN (Liu et al., 2019a) pre-trained on these other attributes (Liu et al., 2019b). Fig. 6 shows the resulting semantic Shapley values explaining the model's prediction of $y = 0$ for an individual in the data. The "unlabelled" bar represents remaining factors of variation not captured by labelled attributes.

Globally, the model makes significant use of age, eyeglasses, and blond hair. This reflects stereotypical western-cultural conceptions of beauty, not surprising in a database of celebrities. The attribute with the largest influence is gender, a result of bias in the data set: women are more than twice as likely to be labelled attractive as men. Locally, the subject's gender (male) thus increases the tendency for the model to predict $y = 0$. The subject's age (young) receives a negative local Shapley value, as this correlates instead with $y = 1$ (opposite the model's prediction) in the data. Hair colour, smile, and eyebrows also played a role in the model prediction for this individual.

Interestingly, our explanation captures dependence on features barely perceptible to the human eye. For example, while traversals of the "young" attribute are hardly noticeable for this individual, this feature significantly impacts the model prediction. Model evaluations on traversals of this feature confirm that this is a genuine pattern exploited by the model and not an aberration in the explanation. Figs. 9 – 12 in App. A.6 provide explicit comparisons between Shapley explanations and model evaluations over a latent-space grid, as well as additional validation of our CelebA experiment.

## 4 RELATED WORK

In this work, we have focused on explaining model predictions in terms of semantic latent features rather than the model's raw inputs. In related work, natural language processing is leveraged to produce interpretable model explanations textually (Hendricks et al., 2018a;b). Other methods (Kim et al., 2018; Zhou et al., 2018) learn to associate patterns of neuron activations with seman-

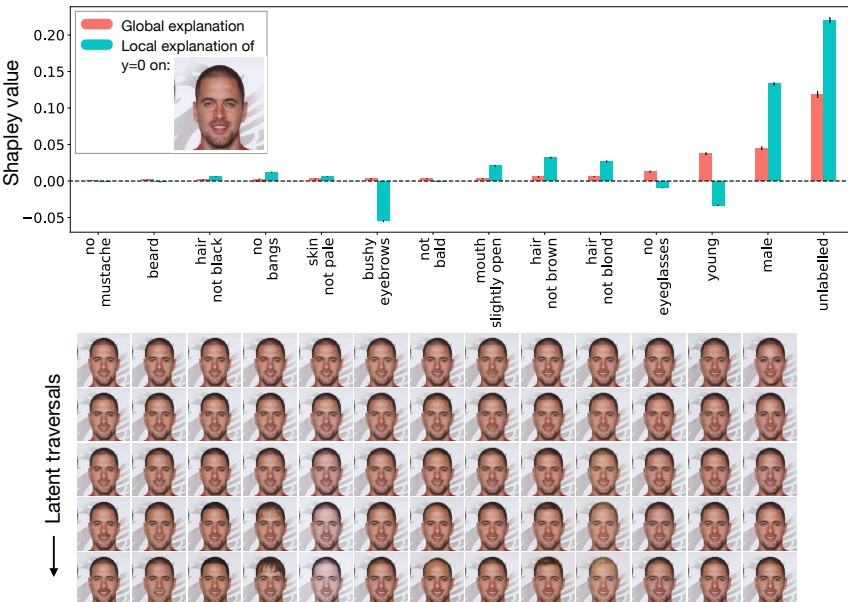

Figure 6: Semantic explanations of a model predicting the attractiveness label in CelebA.

tic concepts. However, each of these requires large sets of annotated data for training – a barrier to widespread application – whereas we offer both unsupervised and analytic options for semantic explainability in our approach. Orthogonal efforts exist to train high-capacity models that are intrinsically interpretable (Alvarez-Melis & Jaakkola, 2018; Chen et al., 2019).

Other related works leverage generative models to produce explanations. Several methods generate counterfactual model inputs (e.g. images) with the features that led to the model prediction accentuated (Singla et al., 2020; Samangouei et al., 2018; Arrieta & Ser, 2020). In contrast to our work, such explanations are pixel-based. Techniques developed by Liu et al. (2019c) generate counterfactual images that cross class boundaries while remaining nearby the original image. When employed using a disentangled generative model, such methods can assign importance to latent features; however, they aim to minimise an ad-hoc pixel-based distance metric, whereas semantic latent features generally control large-scale changes in an image.

A recent workshop (Singh et al., 2020) showed that a change of basis in model explanations can offer insights in cosmology applications. Complementary to this work, we develop this idea in general, offer several alternatives for the basis change, and benchmark with extensive experiments.

Recently released work-in-progress (Wang et al., 2020) studies the dependence of adversarially robust models on the frequency modes of an image's Fourier transform. While taking a somewhat different approach (e.g. explaining based on "Occluded Frequencies") the study finds results generally consistent with ours in Sec. 3.1: adversarial sensitivity is primarily a high-frequency phenomenon.

## 5 CONCLUSION

In this work, we introduced an approach to model explainability on high-dimensional data, in which explanations are articulated in terms of a digestible set of semantic latent features. We adapted the Shapley paradigm to this task, in order to attribute a model's prediction to the latent features underlying its input data. These two developments form a principled, flexible framework for human-interpretable explainability on complex models. To demonstrate its flexibility, we highlighted Fourier transforms, latent disentanglement, and image-to-image translation as options for the semantic representation that offer varying levels of user control. We benchmarked our method on synthetic data, where the underlying latent features are controlled, and demonstrated its effectiveness in an extensive set of experiments using off-the-shelf pretrained models. We hope this framework will find wide applicability and offer practitioners a new way to probe their models.

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

## A  DETAILS OF EXPERIMENTS AND ADDITIONAL RESULTS

Below we provide numerical details and supplementary results for the experiments presented in this paper. For each experiment, we describe both the semantic representation employed as well as the Shapley calculation and its statistical uncertainty.

First we mention the pixel-based explanations that appear in Figs. 1 and 4 (and Fig. 11 below). We produced these using the Captum PyTorch package [`github.com/pytorch/captum`]. We refer the reader to the package documentation for additional details regarding our chosen explanation methods. Explanations using integrated gradients, occlusion, and feature ablation were calculated using a zero-valued baseline.

### A.1  CIFAR-10

For our experiments on CIFAR-10 (Krizhevsky et al., 2009), we explained the predictions of pre-trained robust and non-robust ResNet-50's that are publicly available (Engstrom et al., 2019). The robust model in particular was adversarially (pre-)trained with respect to $\ell_\infty$ perturbations of norm $\varepsilon = 8/255$. For the semantic representation, we used the FFT (Cooley & Tukey, 1965) functionality of NumPy (Oliphant, 2006).

We computed local Shapley values according to Sec. 2.1, using 10k Monte-Carlo samples from the validation set to estimate the local Shapley value for each pixel. The colour plots in Fig. 3 display the means of these Monte-Carlo estimations. (We treated the RGB components of each Fourier mode as a single feature $z_k$ in these calculations. We similarly grouped together complex-conjugate modes, $z_k$ and $z_{k^*}$, as these provide redundant information for real-valued images.) The global Shapley values were similarly computed with 10k samples.

To produce the 1d-comparison plots of Fig. 3, we aggregated Shapley values from the colour plots, binning Shapley values according to the $\ell_2$ norm of the corresponding 2d-frequencies. The shaded bands around the curves display the standard error of the mean in the Monte Carlo sampling discussed above. (These uncertainty bands are very narrow.)

In the final column of Fig. 3, we explain model predictions on adversarially perturbed data. We computed these perturbations according to Engstrom et al. (2019). In particular, we performed projected gradient descent with respect to an $\ell_\infty$ norm of $\varepsilon = 8/255$, using a step size of 0.01 and 100 iterations towards a targeted, randomly-drawn incorrect class. Each (robust and non-robust) model was evaluated on a separate set of direct adversarial attacks.

### A.2  RESTRICTED IMAGENET

Here we present supplementary results on Restricted ImageNet (Engstrom et al., 2019), confirming that the trends found on CIFAR-10 also exist for higher-dimensional images. In particular, we explain the predictions of pre-trained robust and non-robust ResNet-50's (Engstrom et al., 2019); the robust model was adversarially trained with $\ell_\infty$ perturbations of norm $\varepsilon = 4/255$.

In Fig. 7 we present results analogous to those in Fig. 3. In the first row, we show frequency-based Shapley values for each model's prediction on a local image. There we see that the robust model primarily depends on low-frequency information in the image. By contrast, the non-robust model is sensitive to higher-frequency features. This sensitivity leads to the false classification of the adversarially perturbed image of the primate as an insect. In the second row, we show that this trend persists across the data set: the global Shapley values aggregated over many images display trends similar to those just described for an individual image.

For this calculation, we treated all the Fourier modes that fall into the same frequency bin as a single feature in the Shapley calculation. This reduced the $224 \times 224$ Fourier modes down to 25

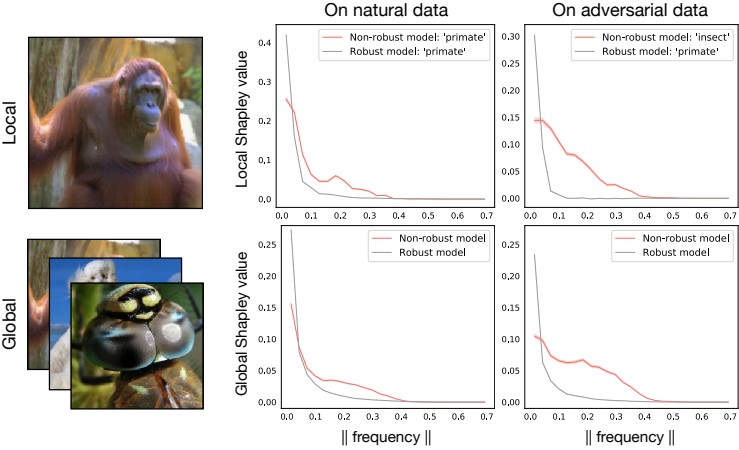

Figure 7: Frequency-based explanations on Restricted ImageNet, providing further evidence that adversarial sensitivity is linked to high-frequency modes.

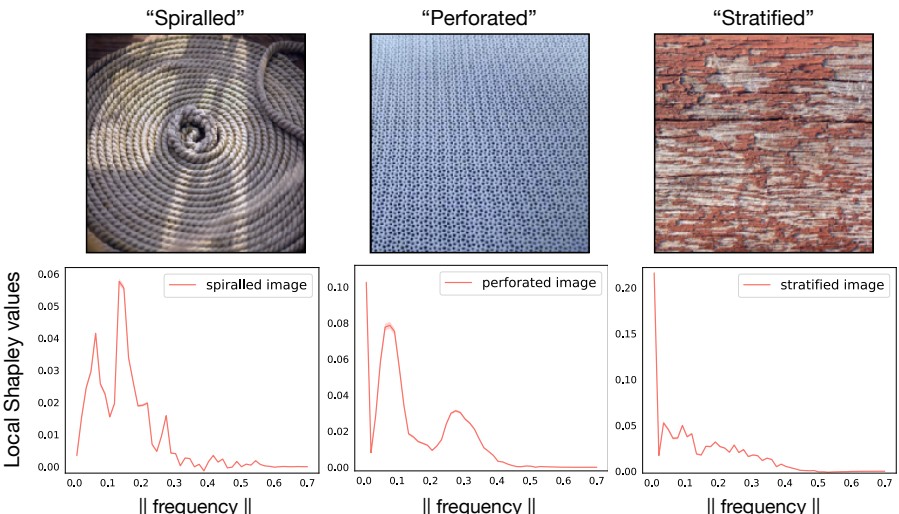

Figure 8: Additional frequency-based explanations on the Describable Textures data set.

aggregate features, thus enabling a very tractable calculation that preserves the primary information-of-interest in the explanation. All other numerical details of our experiment on Restricted ImageNet are identical to those on CIFAR-10, but replacing $8/255 \rightarrow 4/255$ for the adversarial perturbations.

### A.3 DESCRIBABLE TEXTURES

For Describable Textures (Cimpoi et al., 2014), we explained a ResNet-101 (He et al., 2016) trained to predict the 47 textures classes. Our model was optimised using Adam and obtained a top-1 accuracy of 53.5% and a top-5 accuracy of 83.1% on a held-out test set.

Explanations in Fig. 4 were computed according to the same procedure outlined in App. A.2. In particular, 10k Monte Carlo samples were used to estimate the local Shapley value in each frequency bin, and narrow uncertainty bands display the standard error of the mean in each bin. Fig. 8 provides additional local explanations of the model on randomly-drawn "spiralled", "perforated", and "stratified" images. These show clear dependence on specific high-frequency patterns in the images.

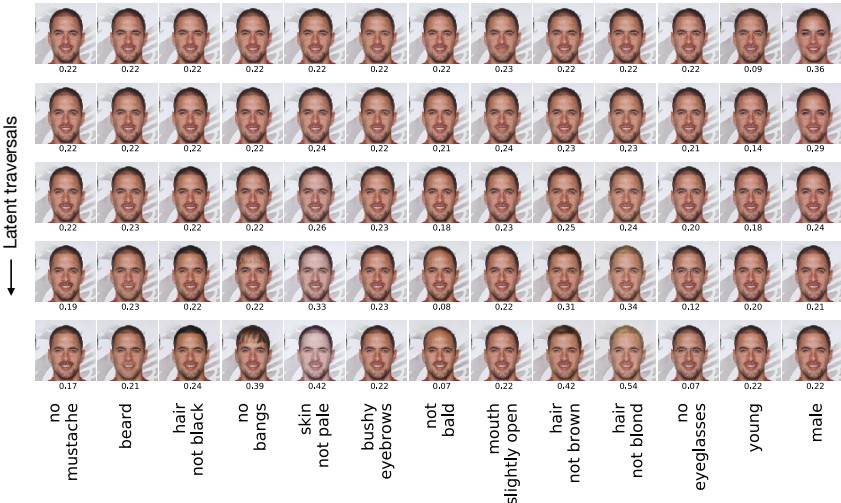

Figure 9: Model dependence on each labelled attribute in CelebA. Below each image in the latent traversals, we show the model's corresponding output. The model was trained to predict the "attractive" label in the data. Its dependence here is consistent with the Shapley explanation in Fig. 6.

## A.4 DSPRITES

For our experiments on dSprites (Matthey et al., 2017), we explained the output of a rules-based model. The model counts the number of white pixels in the top half of the image in order to classify images into 6 classes. The white-pixel-counts naturally fall into 6 clusters because the synthetic dSprites images are generated according to 6 underlying scales.

For the semantic representation, we employed a $\beta$-TCVAE (Burgess et al., 2018), obtaining a publicly available pre-trained network (Dubois, 2019). We applied a rotation matrix to the latent representation to align translation-dimensions with the horizontal and vertical axes.

We computed explanations for Fig. 5a according to Sec. 2.1. Each Shapley value was estimated using 10k Monte Carlo samples: each bar heights represents the mean, and each error bar displays the standard error of the mean.

## A.5 MNIST

For our experiments on MNIST (LeCun & Cortes, 2010), we explained the output of a rules-based model that checks for the presence of a geometric hole in an image. Such a hole exists if there is a black pixel that is not path-connected to the image's perimeter, restricting to paths that only pass through other black pixels. For example, "0" has a hole, because the black pixels at the centre are disconnected from the black pixels near the perimeter, but "1" does not. In practice, we detected holes using SciPy's `binary_fill_holes` function.

For the semantic representation, we employed a JointVAE (Dupont, 2018), using the pre-trained network available at the paper's associated repository. Explanations in Fig. 5b were computed using 10k Monte Carlo samples to estimate the means and standard errors.

## A.6 CELEBA

For our experiments on CelebA (Liu et al., 2015), we pre-processed the data according to the procedure described by Liu et al. (2019a). This entails cropping the central $170 \times 170$ region of the images and using a bicubic linear interpolation to resize them to $128 \times 128$. We explained the predictions of a CNN trained to predict the "attractive" label, which exhibits a $51 : 49$ class balance in the data. Our CNN (with 3 convolutional and 2 fully-connected layers) was optimised using Adam and the cross-entropy loss to achieve 78.7% accuracy on a held-out test set.

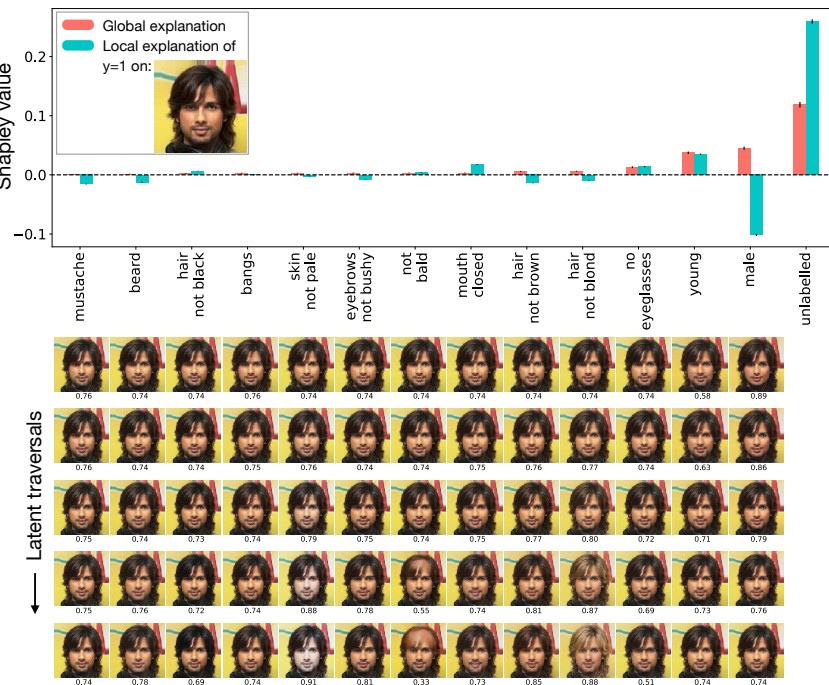

Figure 10: Semantic Shapley explanation of a model predicting the "attractive" label in CelebA. Additional local explanation to complement the one shown in Fig. 6 of the main text.

For the semantic representation, we employed an STGAN (Liu et al., 2019a), using the pre-trained network available at the paper's associated repository. This network was trained to selectively modify the labelled attributes listed in Fig. 6. Our Shapley explanations also include an "unlabelled" dimension: this is meant to represent all remaining factors of variation distinguishing one image from another that are not captured by the labelled attributes. In practice, this unlabelled dimension is simply the image's index in the data set. To vary an image along this unlabelled dimension, one simply draws other images from the data set while using the STGAN to hold their labelled attributes – hair colour, age, gender, etc. – fixed. Shapley values for CelebA were computed using 5k Monte Carlo samples to estimate the mean and standard error for each feature.

In Fig. 9, we provide results that validate the hierarchy of Shapley values that appears in Fig. 6. Below each image in the latent traversals of Fig. 9, we display the CNN's predicted probability that $y = 1$. This shows that the model indeed depends strongly on the rightmost features while remaining relatively insensitive to the leftmost features. While the Shapley explanation of the CNN's prediction also takes into account feature interactions that are not shown here, the model evaluations in Fig. 6 provide a qualitative check on the Shapley values.

Fig. 10 provides an additional explanation of the CNN evaluated on a different image in the data set. The model predicts $y = 1$ for this individual, and this is attributed in large part to the unlabelled factors of variation. The subject's gender (male) anti-correlates with the model's prediction (attractive), while his age (young) is a positive predictor of the model's behaviour.

As a further sanity check on our CelebA explanations, we present supplementary results here explaining a model that predicts the "mouth slightly open" label, which exhibits a $48:52$ class balance in the data. Training our CNN architecture on this task resulted in 91.8% accuracy on the held-out test set. The pixel-based explanations of this classifier are shown in Fig. 11. Since (in contrast to the "attractive" label) this is a visually-localised prediction task, the pixel-based explanations are sensible, highlighting the region surrounding the subject's mouth. Furthermore, the semantic Shapley explanation of this classifier (Fig. 12) attributes the prediction primarily to the "mouth slightly open" latent feature. Since this is indeed the label that the model was trained to predict, this validates our setup for computing semantic explanations on CelebA.

Figure 11: Pixel-based explanations of classifier that predicts "mouth slightly open" label in CelebA.

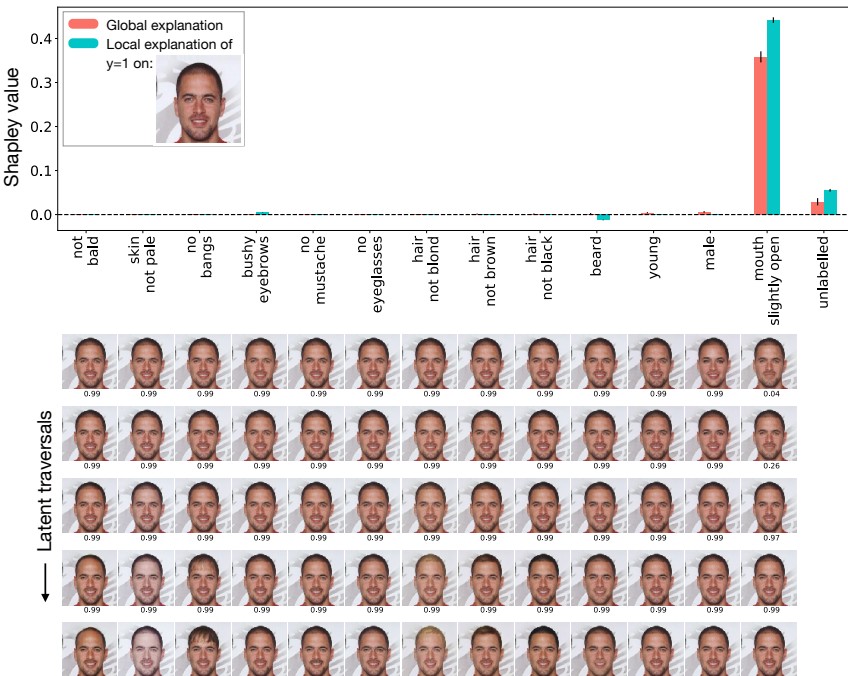

Figure 12: Semantic explanations of classifier that predicts "mouth slightly open" in CelebA.

