# OpenReview forum: "Human-interpretable model explainability on high-dimensional data"
_ICLR.cc/2021/Conference — Reject_

### Official Review · AnonReviewer2 · 2020-10-28
**Interesting direction but feels preliminary**

**Rating:** 4
**Confidence:** 3

**Review:**

Summary: This paper develops a Shapley value approach to explanation that uses low-dimensional latent features to explain the original input. In high-dimensional settings, Shapley values can be computationally intractable; as such, the authors adapt the characteristic (aka value) function $v(\cdot)$ to consider coalitions defined in latent space. This helps ensure that only feasible and plausible (i.e., semantically meaningful) perturbations are made. They consider three different latent encodings: Fourier Transforms, Disentanglement, and image-to-image translation (to isolate factors of variation).

Strengths
- The paper tackles a difficult problem in practice: applying Shapley explanations to high dimensional data. The authors do a fine job of motivating this problem and their subsequent solution.
- The paper is quite thorough in its experimentation as it experiments with CIFAR-10, Describable Textures, dSprites, CelebA, and MNIST.

Weaknesses
- I would have strongly preferred to see a computational analysis of how much more tractable moving to latent space makes the Shapley value. It may be an exponential time reduction if z is chosen to be log(d) for example.
-- Is there a sensible algorithm (or heuristic) for selecting the number of latent factors when encoding? Perhaps there is a tradeoff between the size of z and the explanation quality...

Questions
- Can you comment on if the semantic space needs to be the same space that the discriminative model itself operates in? If you just train an end-to-end model, f, on (x,y) pairs, how can we be sure that the generative, latent factors captured by z are actually used by f?
- Can you comment on the relationship between the Shapley values directly on the input data versus the Shapley values on the latent factors, if any?

---

> ### Author Response · Authors · 2020-11-17
> **Response to Reviewer 2**
>
>
> We thank the reviewer for his review and are glad that he found our approach interesting.
>
> *"I would have strongly preferred to see a computational analysis of how much more tractable moving to latent space makes the Shapley value. It may be an exponential time reduction if z is chosen to be log(d) for example."*
>
> Reply: Although such experiments would without a doubt add value to the paper, at time of writing, we felt that the primary advantage of our method is not it’s computational tractability but rather that it allows for explanation in terms of semantic features such as hair colour, age etc.
>
> *"Is there a sensible algorithm (or heuristic) for selecting the number of latent factors when encoding? Perhaps there is a tradeoff between the size of z and the explanation quality…"*
>
> Reply: In the unsupervised experiments, the number of latent factors is a hyperparameter that can be tuned.
>
> *"Can you comment on if the semantic space needs to be the same space that the discriminative model itself operates in? If you just train an end-to-end model, f, on (x,y) pairs, how can we be sure that the generative, latent factors captured by z are actually used by f?"*
>
> Reply: The discriminative model operates on x. As we perform blackbox model-agnostic explainability, we are not concerned with whether the discriminative model _actually uses_ the latent variables z. Instead, our method provides a way to describe the model's observed behaviour (which undoubtedly relies on very complex internal dynamics) to human users in terms of these more intuitive latent variables.
>
> *"Can you comment on the relationship between the Shapley values directly on the input data versus the Shapley values on the latent factors, if any?"*
>
> Reply: Applying the Shapley values directly on data would generate an explanation in which each pixel was attributed a score quantifying its importance to the model prediction. Applying the Shapley values on latent factors attributes a score to each latent features importance. Because each latent feature is designed to encode a semantic concept such scores can then be attributed to the associated semantic concept.

---

### Official Review · AnonReviewer3 · 2020-10-29
**Natural, useful idea**

**Rating:** 7
**Confidence:** 3

**Review:**

The paper describes a technique for interpreting the results of a neural net in human-readable terms.

The basic idea is natural and simple: (1) find a small number of coordinates for the feature space that correspond to human-interpretable concepts; (2) use Shapley values to assign credit to these coordinates.

Strong points: While there is work related to this idea (which the authors cite appropriately), the overall technique is new. The authors do a good job of backing up the theoretical idea with a set of experiments that seem to produce useful results. (The adversarial example scenario and the celebrity data set use case were especially easy to understand.) The exposition is also very clear.

Weak points: Shapley values are not the only way to exploit human-interpretable coordinates (although they especially are made feasible by the reduction in number of features). The paper might be even stronger if it compared other methods. However, I wouldn't reject the paper on this basis.

I recommend acceptance. This is a solid paper and is likely to be well-cited.

Additional feedback / suggestions for improvement:

* I found the math notation a little confusing. There are three general types of objects: (a) the different spaces where representations live (feature space vs. the interpretable space), (b) elements of those spaces, and (c) functions between those spaces. Certain symbols seemed to do double duty, for instance $\tilde{x}$ seemed to float between these meanings. I'd recommend clear distinctions and contrasting notation for each type of object.

* The authors talk about an "approximate inverse" $\tilde{x}$. It's not really an inverse, maybe more like a pseudoinverse (or maybe a left or right inverse, depending on notation). Perhaps consider alternate terminology.

* When $\tilde{x}$ is differentiable, one could immediately apply other interpretability techniques besides Shapley values (for instance, take a gradient of the class score with respect to the interpretable coordinate space). Ideally, it would be nice to see how these look, so we could compare how much of the clarity of results is due to good coordinates, and how much to use of Shapley values. This would be particularly straightforward in the first Fourier transform example.

---

> ### Author Response · Authors · 2020-11-17
> **Response to Reviewer 3**
>
> We thank the reviewer for taking the time to review the paper and for the constructive comments which will help further improve the paper. We are glad that the reviewer appreciated the paper.
>
> *"Shapley values are not the only way to exploit human-interpretable coordinates (although they especially are made feasible by the reduction in number of features). The paper might be even stronger if it compared other methods. However, I wouldn't reject the paper on this basis.*"
>
> Reply: We agree with the reviewer that different - and potentially useful - explanations can be obtained by replacing Shapley values with other feature attribution methods. However, we feel that Shapley values are particularly well suited here. To recapitulate why. Traditionally Shapley values suffer from two weaknesses. 1) The amount of samples required for generating an explanation scales aggressively with dimensionality. 2) When features are correlated, perturbing features, as done in the shapley calculation, breaks correlations in the dataset and yields non-realistic perturbed data. Shapley values are thus particularly well suited for applications on disentangled models as these issues are severely reduced since the number of features is drastically reduced and features are by design uncorrelated.

---

### Official Review · AnonReviewer4 · 2020-10-30
**Interesting method but no comparison to other concept-level explanations**

**Rating:** 3
**Confidence:** 4

**Review:**

Summary: I think that what is presented is a promising method for human interpretation of high-dimensional models. However, the experiments feel too much like toy examples, without rigorous attempts to validate the resulting feature attributions or to compare to other ways of getting concept-level attributions.

**Update**: After reading the rebuttal, I am maintaining my previous score. I believe that, given the related work reviewers have mentioned, the paper most likely requires some experimental comparisons with these other methods. The rebuttal arguments are good, but not convincing enough that I can believe in this method's superiority without seeing a comparison with, e.g., CaCE.

Objective: Explain the output of high-dimensional ML models in a human-interpretable way by using Shapley values on semantically meaningful latent features.

Strengths:
* This is an extremely important problem; the shortcomings of e.g. pixel-based methods for images are important and well-noted.
* The proposed method is a promising way to attribute to (and in some cases, learn) semantically meaningful latent features.
* dSprites serves as a good ground truth where the generative process is known, and serves to validate some of the human-interpretable patterns found.

Weaknesses:
My overall concern is that while the experimental results are interesting, they are not thorough and don't demonstrate the marginal value of this method relative to other possible methods in the space; for example, concept bottleneck or causal concept effect models.
* I didn't see a reference to Koh et al's "Concept Bottleneck Models" (https://arxiv.org/abs/2007.04612), which do supervised learning of semantic concepts then train a classifier on top of those concepts, which is very similar/relevant work.
* Similarly I didn't see a reference to Goyal et al's "Explaining Classifiers with Causal Concept Effect (CaCE)", which trains a VAE to learn a meaningful latent space and report causal effects of modifications to the latent variables.
The existence of this prior work sharpens some specific questions I had when reading the paper:
* For the non-Fourier tasks, why use a VAE latent space? It's impossible to know that VAE latent dimensions (when they do look like they correspond to a concept) correspond *only* to that concept. The bar charts stating "most importance goes to vertical position/digit identity" seem overconfident, since that is only our guess of what the latent dimension means. In contrast, a concept bottleneck model has dimensions with clear meaning (they were trained with supervision). This is acknowledged to some degree in the text.
* Also, for the non-Fourier tasks, why use the Shapley framework? CaCE is a very similar approach, using a VAE to get explanations in terms of an interpretable latent space ,but doesn't use Shapley. What are the pros and cons of each approach? Also, given a meaningful latent space, why not use gradient-based methods (i.e., integrated gradients) to attribute to it?
Overall, I don't think this method needs to blow the others out of the water; there are legitimate counterpoints to be made -- for example, concept bottleneck models require concept labels, though I believe such labels are available in the CelebA experiments. But a thoughtful discussion of the relationship of this work to methods like concept bottlenecks or CaCE is essential -- and not present in the current version.
Other weaknesses:
* The quality of the resulting explanations is not assessed rigorously. The fact that the dSprites example picks up on vertical position and the MNIST example picks up on digit value is useful, but many interpretability papers look more exhaustively, across the dataset, at what happens to the model output when the features reported as important are at least perturbed or ablated, or if a model is retrained with the features altered. I would particularly like to see such a comparison with concept bottleneck and/or CaCE: if this method (Shapley with VAE) is the right way to do things, perhaps it will outperform the other methods at predicting how much certain concept shifts will affect model output or model retraining.
* The Fourier examples are interesting but feel a bit like toy examples to me because they are a specific case with a convenient invertible feature map and mostly recapitulate known properties of adversarial examples (they rely on high-frequency patterns).
* Minor comment: The "landscape of semantic representations" could be shortened, potentially leaving room for more comparisons to other methods/benchmarks of the attributions.

---

> ### Author Response · Authors · 2020-11-17
> **Response to Reviewer 4**
>
> We thank the reviewer for his thoughtful review
>
>
> *"I didn't see a reference to Koh et al's "Concept Bottleneck Models" (https://arxiv.org/abs/2007.04612), which do supervised learning of semantic concepts then train a classifier on top of those concepts, which is very similar/relevant work. Similarly I didn't see a reference to Goyal et al's "Explaining Classifiers with Causal Concept Effect (CaCE)", which trains a VAE to learn a meaningful latent space and report causal effects of modifications to the latent variables."*
>
> Reply: We thank the reviewer for the link to these relevant papers that we had overlooked. Especially the CaCE paper. We will update our literature section.
>
>
> *"For the non-Fourier tasks, why use a VAE latent space? It's impossible to know that VAE latent dimensions (when they do look like they correspond to a concept) correspond only to that concept. The bar charts stating "most importance goes to vertical position/digit identity" seem overconfident, since that is only our guess of what the latent dimension means. In contrast, a concept bottleneck model has dimensions with clear meaning (they were trained with supervision). This is acknowledged to some degree in the text."*
>
> Reply: Concept bottleneck models provide a framework for generating interpretable models but do not allow for explaining existing non-interpretable models. As such they are not applicable in many situations where one would want a model explanation. For example, when auditing an existing deployed model. It is also worth pointing out that such a comparison is somewhat unfair since Concept-Bottleneck Models require supervised labels, while the MNIST experiment was done fully unsupervised. Given access to appropriate labels, our method could be applied on a latent with labels disentangled supervisedly -  as is done in the CelebA experiment.
>
>
> *"Also, for the non-Fourier tasks, why use the Shapley framework? CaCE is a very similar approach, using a VAE to get explanations in terms of an interpretable latent space ,but doesn't use Shapley. What are the pros and cons of each approach? Also, given a meaningful latent space, why not use gradient-based methods (i.e., integrated gradients) to attribute to it? Overall, I don't think this method needs to blow the others out of the water; there are legitimate counterpoints to be made -- for example, concept bottleneck models require concept labels, though I believe such labels are available in the CelebA experiments. But a thoughtful discussion of the relationship of this work to methods like concept bottlenecks or CaCE is essential -- and not present in the current version. "*
>
> Reply: Both CaCe and Shapley values measure feature importance for a model’s prediction by switching the considered feature on and off. However, the methods diverge in how the remaining features are treated. In CaCE remaining features are kept at their current values. This leads to correlation amongst features not being accounted for when generating explanations. To give a concrete example, if a task consisted in classifying images into two classes, where one class contains images of red squares, green squares and red triangles and another class contains images of green triangles. Applying CaCE to explain predictions of a perfect classifier on an image of a red square would incorrectly yield an explanation in which neither colour nor shape were judged important. This is because changing either one of these features would not change the model output. Shapley values expressly consider all combinations of features to account for such cross-feature effects. Such types of behaviors are highlighted in our DSprites experiment where feature correlation is important for the model outputs.

---

### Official Review · AnonReviewer1 · 2020-11-01
**Interesting approach but insufficient comparison with previous works and concern of faithfulness**

**Rating:** 5
**Confidence:** 4

**Review:**

Summary:

The paper proposes an approach to generate semantic explanations for high-dimensional data. The proposed approach consists of two modules -- the first module transforms the high-dimensional raw data into lower-dimensional semantic latent space and the second module applies Shapely explainability to this lower-dimensional latent space to generate explanations in terms of semantic concepts. The approach has been applied to six different datasets.

—————————————————————————————————————————————————————————————


Strengths:

S1) The paper is very well written and easy to understand.

S2) The proposed approach is modular consisting of 2 modules. This allows the flexibility to replace each module with more efficient/advantageous methods as research progresses in future.

S3) The paper proposes three different ways of transforming high-dimensional raw data to lower-dimensional latent space, over which Shapely explainability can be applied to generate explanations in terms of semantic concepts.

S4) The paper shows results for six different datasets.

—————————————————————————————————————————————————————————————


Weaknesses:

W1) The authors should compare their approach (methodologically as well as experimentally) to other concept-based explanations for high-dimensional data such as (Kim et al., 2018), (Ghorbani et al., 2019) and (Goyal et al., 2019). The related work claims that (Kim et al., 2018) requires large sets of annotated data. I disagree. (Kim et al., 2018) only requires a few images describing the concept you want to measure the importance of. This is significantly less than the number of annotations required in the image-to-image translation experiment in the paper where the complete dataset needs to be annotated. In addition, (Kim et al., 2018) allows the flexibility to consider any given semantic concept for explanation while the proposed approach is limited either to semantic concepts captured by frequency information, or to semantic concepts automatically discovered by representation learning, or to concepts annotated in the complete dataset. (Ghorbani et al., 2019) also overcomes the issue of needing annotations by discovering useful concepts from the data itself. What advantages does the proposed approach offer over these existing methods?

W2) Faithfulness of the explanations with the pretrained classifier. The methods of disentangled representation and image-to-image translation require training another network to learn a lower-dimensional representation. This runs the risk of encoding some biases of its own. If we find some concerns with the explanations, we cannot infer if the concerns are with the trained classifier or the newly trained network, potentially making the explanations useless.

W3) In the 2-module approach proposed in the paper, the second module can theoretically be any explainability approach for low-dimensional data. What is the reason that the authors decide to use Shapely instead of other works such as (Breiman, 2001) or (Ribeiro et al., 2016)?

W4) Among the three ways of transforming the high-dimensional data to low-dimensional latent space, what criteria should be used by a user to decide which method to use? Or, in other words, what are the advantages and disadvantages of each of these methods which might make them more or less suitable for certain tasks/datasets/applications?

W5) The paper uses the phrase “human-interpretable explainability”. What other type of explainability could be possible if it’s not human-interpretable? I think the paper might benefit with more precise definitions of these terms in the paper.


References mentioned above which are not present in the main paper:

(Ghorbani et al., 2019) Amirata Ghorbani, James Wexler, James Zou, Been Kim. Towards Automatic Concept-based Explanations. NeurIPS 2019.

(Goyal et al., 2019) Yash Goyal, Amir Feder, Uri Shalit, Been Kim. Explaining Classifiers with Causal Concept Effect (CaCE). ArXiv 2019.




—————————————————————————————————————————————————————————————— ——————————————————————————————————————————————————————————————

Update after rebuttal: I thank the authors for their responses to all my questions. However, I believe that these answers need to be justified experimentally in order for the paper’s contributions to be significant for acceptance. In particular, I still have two major concerns. 1) the faithfulness of the proposed approach. I think that the authors’ answer that their method is less at risk to biases than other methods needs to be demonstrated with at least a simple experiment. 2) Shapely values over other methods. I think the authors need to back up their argument for using Shapely value explanations over other methods by comparing experimentally with other methods such as CaCE or even raw gradients. In addition, I think the paper would benefit a lot by including a significant discussion on the advantages and disadvantages of each of the three ways of transforming the high-dimensional data to low-dimensional latent space which might make them more or less suitable for certain tasks/datasets/applications. Because of these concerns, I am keeping my original rating.

---

> ### Author Response · Authors · 2020-11-17
> **Response to Reviewer 1**
>
> We thank the reviewer for reading our paper in depth and taking the time to write a thoughtful and detailed review.
>
> *“The related work claims that (Kim et al., 2018) requires large sets of annotated data. “*
>
> Reply: Although we made the above statement in reference to the unsupervised disentanglement case, we agree with the reviewer that the above listed concept-based explanations require fewer samples than our method applied to supervised disentanglement.
>
> *"What advantages does the proposed approach offer over these existing methods?"*
>
> Reply: Perceptual similarity methods require attributing directions in the neural network representation to concepts. Explanations created using such methods rely on the somewhat fragile assumptions that concepts are well represented by a single linear direction in the neural network embedding space, and that these directions can be correctly teased out from a limited sample of images. For example it can be difficult to ascertain that a model has learned the concept of stripes and that other black and white patches would not trigger the model, or that a complex concept like a car can be well represented using only one unique direction in the latent space as would be required by these methods.
>
> Our method offers a more rigorous treatment of model explainability. Shapley values provide a well-studied theoretically motivated framework for feature attribution. Because our explanations rely only on counterfactually generated data, our explanations do not involve any black-box components. Indeed, one can look at the counterfactually generated data to ascertain that it captures desired concepts.
>
> *"Faithfulness of the explanations with the pretrained classifier. The methods of disentangled representation and image-to-image translation require training another network to learn a lower-dimensional representation. This runs the risk of encoding some biases of its own. If we find some concerns with the explanations, we cannot infer if the concerns are with the trained classifier or the newly trained network, potentially making the explanations useless.*"
>
> Reply: We agree with this assessment and have cautioned in the paper that our interpretations can carry biases if the latent directions do not coincide with the concepts we care about. However, as stated above, we feel that our approach is less at risk to these biases than other methods, since our explanation is based on artificial data which can be double-checked for biases.
>
> *"In the 2-module approach proposed in the paper, the second module can theoretically be any explainability approach for low-dimensional data. What is the reason that the authors decide to use Shapely instead of other works such as (Breiman, 2001) or (Ribeiro et al., 2016)?"*
>
> Reply: All attribution methods have their strengths and weaknesses. Shapley values provide a comprehensive causally motivated approach to feature attribution that accounts for feature interactions. However when applied on raw features this comes with two weaknesses. 1) The amount of samples required for generating an explanation scales aggressively with dimensionality. 2) When features are correlated, perturbing features, as done in the shapley calculation, breaks correlations in the dataset and yields non-realistic perturbed data. Shapley values are thus particularly well suited for applications on disentangled models as these issues are severely reduced since the number of features is drastically reduced and features are by design uncorrelated.
>
> *"Among the three ways of transforming the high-dimensional data to low-dimensional latent space, what criteria should be used by a user to decide which method to use? Or, in other words, what are the advantages and disadvantages of each of these methods which might make them more or less suitable for certain tasks/datasets/applications?"*
>
> Reply: In our opinion, all methods of transforming the dataset have their practical use cases. Unsupervised representation allows for getting a quick breakdown of important factors of variation without needing any annotations. When feasible, supervised transformations allow for a principle and very precise breakdown of how much each feature matters.
>
> *"The paper uses the phrase “human-interpretable explainability”. What other type of explainability could be possible if it’s not human-interpretable? I think the paper might benefit with more precise definitions of these terms in the paper"*
>
> Reply: In our choice of vocabulary, we wished to express how saliency methods applied to raw features are incapable of effectively conveying some important factors of variation such as relative position, colour that can be distinguished using an appropriate basis as is done through our framework.

---

### Decision · Program_Chairs · 2021-01-07
**Final Decision**

**Decision:**

Reject

**Comment:**

This paper introduces an approach to model explainability on high-dimensional data by: (1) first mapping inputs to a smaller set of intelligible latent features, and then (2) applying the Shapley method to this set of latent features. Several methods are considered for (1), and empirical results are examined across several settings.

Reviewers were mixed in their views - one reviewer was in favor of acceptance, and three were against.

Some concerns were addressed by authors in the discussion period but remaining issues include:
concerns over the faithfulness of the approach;
missing comparisons to other related methods such as CaCE; and
a desire for more in-depth discussion of the pros and cons of the different methods considered for (1).